# MRSA in Humans, Pets and Livestock in Portugal: Where We Came from and Where We Are Going

**DOI:** 10.3390/pathogens11101110

**Published:** 2022-09-27

**Authors:** Vanessa Silva, Andreia Monteiro, José Eduardo Pereira, Luís Maltez, Gilberto Igrejas, Patrícia Poeta

**Affiliations:** 1Microbiology and Antibiotic Resistance Team (MicroART), Department of Veterinary Sciences, University of Trás-os-Montes and Alto Douro (UTAD), 5000-801 Vila Real, Portugal; 2Department of Genetics and Biotechnology, University of Trás-os-Montes and Alto Douro (UTAD), 5000-801 Vila Real, Portugal; 3Functional Genomics and Proteomics Unit, University of Trás-os-Montes and Alto Douro (UTAD), 5000-801 Vila Real, Portugal; 4Associated Laboratory for Green Chemistry (LAQV-REQUIMTE), University NOVA of Lisboa, 2829-516 Caparica, Portugal; 5CECAV—Veterinary and Animal Research Centre, University of Trás-os-Montes and Alto Douro (UTAD), 5000-801 Vila Real, Portugal; 6Associate Laboratory for Animal and Veterinary Science (AL4AnimalS), University of Trás-os-Montes and Alto Douro (UTAD), 5000-801 Vila Real, Portugal

**Keywords:** MRSA, *Staphylococcus aureus*, Portugal, animals, humans

## Abstract

Over the years, molecular typing of methicillin-resistant *S. aureus* (MRSA) has allowed for the identification of endemic MRSA strains and pathogenic strains. After reaching a peak of predominance in a given geographic region, MRSA strains are usually replaced by a new strain. This process is called clonal replacement and is observed worldwide. The worldwide spread of hospital-associated MRSA (HA-MRSA), community-associated MRSA (CA-MRSA) and livestock-associated MRSA (LA-MRSA) clones over the last few decades has allowed this microorganism to be currently considered a pandemic. In Portugal, most HA-MRSA infections are associated with EMRSA-15 (S22-IV), New York/Japan (ST5-II) and Iberian (ST247-I) clones. Regarding the strains found in the community, many of them are frequently associated with the hospital environment, namely the Pediatric, Brazilian and Iberian clones. On the other hand, a strain that is typically found in animals, MRSA clonal complex (CC) 398, has been described in humans as colonizing and causing infections. The ST398 clone is found across all animal species, particularly in farm animals where the economic impact of LA-MRSA infections can have disastrous consequences for industries. In contrast, the EMRSA-15 clone seems to be more related to companion animals. The objective of this review is to better understand the MRSA epidemiology because it is, undoubtedly, an important public health concern that requires more attention, in order to achieve an effective response in all sectors.

## 1. Introduction

*Staphylococcus aureus* persistently or intermittently colonizes the nasal mucosa in approximately 30% of healthy adults [1]. Individuals naturally colonized by *S. aureus* have an increased risk of infection by this bacterium since its presence does not cause a detectable immune response in the host. In addition, these individuals are an important source of transmission of the microorganism to other people, which usually occurs through direct contact with the skin (Lee et al., 2018).

Antimicrobial resistance is a global public health problem that must be addressed by health authorities worldwide. Estimates from the European Union/European Economic Area show that each year more than 670,000 infections are due to antimicrobial-resistant bacteria with approximately 33,000 deaths [2].

The current methicillin-resistant *Staphylococcus aureus* (MRSA) situation in Europe has a prevalence of <5% in 9 European Union countries and >25% for 10 countries in the same region, namely Portugal [2]. In the hospital sector, resistant microorganisms deserve special attention. MRSA is one of the most frequent microorganisms that cause infections associated with the hospital environment [3]. In that same year, *S. aureus* were responsible for 18.5% of pneumonia episodes and 12% of bloodstream infections acquired in intensive care units in European hospitals. Of the *S. aureus* identified, 20% were MRSA [4]. In Portugal, 13.8% of pneumonia cases and 9.7% of bloodstream infections were caused by *S. aureus* strains [4]. The great variability in the rates of infections associated with the hospital environment in the various European countries is partly related to the different diagnostic methods. However, these data are of great importance as they alert to the need to implement measures to control and prevent the emergence and spread of resistant organisms.

Initially, MRSA infections were predominantly associated with the hospital environment, occurring mostly in hospitalized patients or patients who attended that environment. Therefore, during the 1960s−1970s, they were called HA-MRSA (Gajdács, 2019). However, in the late 1990s, MRSA infections began to appear in the community in people with no contact with the hospital environment and were therefore called CA-MRSA (Turner et al., 2019). These MRSA strains (HA-MRSA and CA-MRSA) belong to distinct genetic lineages and have some differences. HA-MRSA strains are generally more resistant to numerous drugs and have larger Staphylococcal Cassette Chromosome *mec* (SCC*mec*) types I, II and III. In contrast, CA-MRSA strains often have smaller SCC*mec* elements, usually types IV and V, and are not as resistant [5]. However, these strains appear to be more virulent due to the expression of virulence factors that increase their pathogenic potential [6], being one of the most frequent etiological agents of skin and soft tissue infections [7].

HA-MRSA are nosocomial and arise from wounds of infected patients, catheters and prolonged hospitalization, but also from the skin of healthy carriers [8]. In the late 1990s, MRSA infections began to appear in the community in people with no contact with the hospital environment [9]. The community-associated MRSA (CA-MRSA) has become a point of concern since these strains are associated with higher levels of virulence and disseminate fast, thus affecting seemingly healthy individuals [10]. In recent years, numerous MRSA strains have been isolated from different animal species, especially pigs, in many countries and were called livestock-associated MRSA (LA-MRSA) [11,12,13]. However, in most cases, MRSA colonized the animal host asymptomatically [14]. Most LA-MRSA belong to the genetic lineage clonal complex (CC) 398. Strains belonging to this same genetic lineage have been found in people who have direct contact with livestock [12,15,16]. In addition to this, vancomycin resistance has become a major concern within the scientific community. This adaptation arises from the acquisition of accessory components, in this case, the *van*A gene [17], leading to the emergence of Vancomycin-resistant *S. aureus* (VRSA). This genetic adaptation is extremely important since there is a wide dependence on this antibiotic in the treatment of infections caused by MRSA. Vancomycin resistance in *S. aureus* tends to appear after prolonged or repeated periods of treatment with vancomycin, in a phenomenon called heteroresistance, where multiple mutations occur conferring different degrees of resistance to the antibiotic [18].

Currently, there has been an increase in the number of nosocomial infections by CA-MRSA; HA-MRSA clones have been responsible for infections in the community and LA-MRSA clones have been detected in healthy humans and human infections. Apparently, these clones have the ability to cross physical barriers, adapting easily. This change in the epidemiology of MRSA has raised serious concern as it becomes difficult to define the boundary between hospital–community-livestock transition. The purpose of this review is to better understand the MRSA epidemiology in order to achieve better answers in all sectors.

## 2. MRSA Clones

### 2.1. Human-Associated MRSA

The control and prevention of infections by *S. aureus* strains can be carried out through rapid molecular typing that, simultaneously allows us to understand the transmission mechanism of this microorganism. MRSA typing can be performed through phenotypic or molecular characterization. In fact, the identification of endemic MRSA strains and strains responsible for disease outbreaks has become possible due to the improvement in molecular characterization techniques, which have allowed for a greater discrimination of MRSA clones [19]. Multilocus sequence typing (MLST) consists of the analysis of nucleotide sequences of internal fragments of seven housekeeping genes present in *S. aureus*. These genes are highly conserved since they encode enzymes necessary for the metabolism of the bacteria. The sequence of each locus is assigned an allele identification number based on its similarity to known alleles, and the combination of these seven alleles generates a sequence type (ST). When there are single nucleotide polymorphisms in fewer than three genes, the STs are considered closely related, and can be grouped under the same clonal complex (CC) [20]. The *spa*-typing technique is also widely used and it is based on the amplification and sequencing of the 24 bp X polymorphic zone of the *spa* gene, which encodes protein A, containing a variable number of repeats [21]. These repetitions are contrasted with those already known on an online server and, thus, the number associated with that strain of *S. aureus* can be determined.

MRSA strains known today resulted from processes of genetic recombination of pre-existing strains. Throughout this process of evolution, there was a selection of some advantageous characteristics that made these strains successful in a given geographic location [22,23]. Furthermore, the predominant lineage in that geographic location, after reaching a peak of dominance, tends to decline and later disappear, being replaced by a new lineage [24]. This clonal replacement process has been observed worldwide [25]. The acquisition of different types of Staphylococcal Cassette Chromosome *mec* (SCC*mec*) by methicillin-susceptible *S. aureus* (MSSA) strains of different genetic origins underlies the origin of MRSA pandemic clones such as those belonging to CC8: Archaic (ST250-I), Iberian (ST247-I), Brazilian (ST239-III) and USA300 (ST8-IV); and clones belonging to CC5, CC22, CC30 and CC398: New York/Japan (ST5-II), EMRSA-15 (ST22-IV), EMRSA-16 (ST36-II) and LA-MRSA (ST398-V), respectively [19,26].

MRSA infections in Europe in the early 1960s were limited to hospital outbreaks caused primarily by *S. aureus* with the phage 83A (later designated ST250). This clone was designated Archaic and was gradually replaced in the following decades by five other prevalent clones: CC5, CC8, CC22, CC30 and CC45 [6]. Of these, clones CC5 and CC8 are the most prevalent worldwide and comprise several different Sequence Types (ST’s) being widely distributed throughout the world (Table 1) [19].

Currently, Portugal has one of the highest nosocomial prevalence rates in Europe. Most HA-MRSA infections in Portugal are associated with EMRSA-15, the Iberian clone, and New York/Japan clones [29,30]. The Iberian clone was first described in Spain in 1989 and, since then, it has been reported in several countries, namely in Portugal. This, and the Portuguese clone (ST239-III variant) were the most prevalent clones of HA-MRSA from the mid-1980s to the beginning of the following decade [31,32]. The Pediatric clone was also mentioned for the first time in 1992, in a pediatric hospital in Portugal and was later found in Poland, the United States, Argentina and Colombia. Currently, the Clone New York/Japan is the most prevalent HA-MRSA clone in the country [19].

In addition to these, other clones such as the Brazilian (first described in 1992, in Brazil) have been identified worldwide. The New York/Japan clone, in turn, was identified as the dominant MRSA in hospitals in the metropolitan areas of New York, Pennsylvania, New Jersey and Connecticut in the United States, and also in a hospital in Tokyo, Japan [33]. However, in the last decade, there was a major shift in dominant MRSA clones from the New York/Japan to the USA300 clone [33]. In the United Kingdom, EMRSA-15 and EMRSA-16 were identified as endemic in hospitals in that region during the 1990s [34]. The same was true in several other European countries. Currently, EMRSA-15 is the dominant strain in UK hospitals, emphasizing the fact that CA-MRSA clones are widely disseminated in the hospital setting [9].

### 2.2. Animal-Associated MRSA

In addition to humans, MRSA also colonizes and infects animals. MRSA colonization and infection have been reported in a variety of animals, from domestic animals to farm animals. The indiscriminate use of antimicrobial agents in animal production and other agricultural activities has largely contributed to the distribution of MRSA among animals [35], thus, affecting more than 40% of pigs, 20% of cattle and up to 90% of turkey farms as demonstrated by a study carried out in Germany [36]. In Europe, LA-MRSA mostly belongs to the ST398 of the CC398, presenting itself as the largest reservoir of MRSA [37]. First recognized in 2005, LA-MRSA CC398 strains are the main type of MRSA reported in swine internationally and it is conceivable that this strain, which originated in swine, was later dispersed to other animals, such as cattle, horses and poultry [37,38,39,40,41]. While CC398 is the dominant lineage in livestock in Europe, other STs are predominant in other geographic locations. The dominant MRSA in livestock in Asia is CC9 [42]. Nevertheless, a wide range of genetic lineages have been reported in livestock such us CC1, CC5, CC9, CC45, CC97 and CC398 [43]. Figure 1 shows the distribution of MRSA CCs most commonly found in livestock and pets. MRSA strains often carried the *mec*A gene which confers resistance to methicillin but, the divergent *mec*A variant, *mec*C gene, has been increasingly reported in animals since it was first recognized in 2011 [44]. *mec*C-MRSA has been found in humans and animals, including pets, wild animals and livestock [11,45,46]. Most of the *mec*C-MRSA found among livestock belong to CC130 or CC425 [28]. Generally, MRSA clones of pets differ from those in livestock and meat production animals [47]. In cats and dogs, the most predominant MRSA lineage is ST22-IV (EMRSA-15) in Europe while in North America and Japan is ST5 [48,49]. As for horses, the most common lineages are ST8 and ST398 [41,50].

## 3. MRSA in Portugal

### 3.1. MRSA in Humans

Over the years, the clonal strains of MRSA prevalent in humans in Portugal have shifted, as has happened in other countries. Several studies conducted in Portugal between 2000 and 2010 studied the clonal replacement process that took place at a hospital in Oporto over several years. According to the authors, between 1992 and 1993, the Iberian clone (ST247-I) was the dominant clone representing 77% of the isolates [51,52,53]. Apparently, in the middle of that decade, this clone was replaced by the Brazilian clone (ST239-III), a scenario that occurred in several Portuguese hospitals. From that time until the early 2000s this multidrug-resistant clone was considered the dominant clone in this hospital. Subsequently, the Brazilian clone was replaced by the EMRSA-15 (ST22-IV), from 2003 to 2005, which became predominant. At this time, two other MRSA pandemic clones, the New York/Japan (ST5-II) and the EMRSA-16 (ST36-II), were identified in this hospital for the first time (Figure 2).

Espadinha et al. studied the prevalence of MRSA in several Portuguese health institutions and reported that in 2010, about 20% of the MRSA isolates belonged to CC5 (ST105-II, ST125-IVc and ST5-IVc) indicating a possible shift of MRSA clones in Portuguese hospitals. Furthermore, the authors also concluded that the MRSA clones identified in the community were clones typically associated with the hospital environment [54]. MRSA in Portugal has been widely isolated from invasive infections, including bloodstream infections. In a study conducted in 2013 by Faria et al., the EMRSA-15 clone was the most prevalent in bloodstream infections [55]. However, in the same study, a variant of the New York/Japan clone (ST105-II) was the second most frequent clone and the authors proposed that the ST105-MRSA-II clone could replace the EMRSA-15 and be the next clonal wave of MRSA in Portuguese hospitals. However, a more recent study published in 2019 with MRSA from bloodstream infections revealed that EMRSA-15 still remains the most frequent clone showing that no substitution of EMRSA-15 by the New York/Japan variant clone in Portuguese hospitals as previously suggested [30]. Initially, EMRSA-15 clone in Portugal was mainly ascribed to *spa*-types t747, t032, and t2357; however, as EMRSA-15 became the main clone there was an increase in *spa* diversity (Faria et al., 2013). EMRSA-15 clone often carried the SCC*mec* type IV which is smaller and has a lower fitness cost compared with other SCC*mec* types. This feature allows an easier dissemination of this clone [56]. In 2013, Tavares et al. studied a total of 1487 *S. aureus* isolated from patients attending 16 Portuguese health institutions located in different geographic regions of the country. Patients responded to a questionnaire aimed at distinguishing between CA-MRSA and HA-MRSA infections. After collecting the questionnaires, 527 (41.6%) isolates were defined as having a community origin and 740 (58.4%) isolates were classified as having a hospital origin. Of the 527 isolates of community origin, 114 (21.6%) were MRSA, of which 101 (88.6%) belonged to epidemic clones of HA-MRSA and only 13 (11.4%) were CA-MRSA [57].

In a study conducted in Portugal in 2011, 38 *S. aureus* were isolated from skin and soft tissue infections of children admitted to a pediatric hospital and 3 (7.9%) were identified as MRSA. Two of them belong to CC5 and the third to CC80 [58]. The CC5 clones are related to the Pediatric clone, while the CC80 clone is associated to the European CA-MRSA clone. In another study, hospital samples from skin and soft tissue infections were analyzed in two different time periods (1993 and 2010) with the aim of analyzing the clonal nature of MRSA and verifying whether there were any epidemiological changes. The samples were obtained from a hospital in Lisbon and from some health centers in the north, center and south. The results obtained showed that, in 1993, 54 MRSA isolates were identified including the Iberian clone (ST247-I, t008/t051) and the Portuguese clone (ST239-III variant, t421). On the other hand, in the 2010 samples, 180 MRSA isolates were identified, among them the clones EMRSA-15, Pediatric, New York/Japan and Pediatric. In general, these results suggest the dissemination of typical HA-MRSA clones in the community. Staphylococci are also one of the most predominantly found in diabetic foot ulcers (DFU) [59,60]. In Portugal, several studies have investigated the organisms responsible for DFU infections and some have also investigated the frequency and genetic lineages of MRSA from diabetic foot ulcers [29,59,61,62,63,64]. Mottola et al. 2016 performed the molecular characterization of MRSA isolates from DFU infections in diabetic patients and reported that ST22 and ST5 were the most frequent clones. In the study conducted by our research group, multidrug-resistant EMRSA-15 clone was the most frequent clone in DFUs [29,64]. The enormous diversity of isolates found in diabetic foot ulcers may be related to the need for these patients to travel frequently to health centers to perform wound treatment. The isolates frequently found in this pathology include CC5 and CC22, which are the main clones associated with hospitals in Portugal. In a study from Mottola el al., 36% of staphylococci from diabetic foot were multi-drug resistant and the higher resistances were obtained for ciprofloxacin and erythromycin [59].The people with these pathology can easily be a vehicle for the spread of bacterial strains from hospitals to the community and vice versa [59].

Regarding the vancomycin resistance problematic, the first vancomycin-resistant *S. aureus* isolate detected in Europe occurred in a 74-year-old diabetic woman. Subsequently, from the pus resulting from the amputation wound, a methicillin-resistant VRSA and vancomycin-resistant *Enterococcus faecalis* (VRE) were isolated. Both isolates contained the *van*A gene and, in addition to this, the *mec*A gene found in the VRSA strain which was classified as type ST105 and SCC*mec* type II [65]. The results of this study suggest that the most likely hypothesis for the emergence of VRSA is through the transfer of the *van*A gene from the donor VRE strain in Portugal, the first MRSA with intermediate resistance to vancomycin (VISA) was isolated in 2006 from a surgical wound of a patient hospitalized at the orthopedics ward of Hospital de São Marcos–Braga. This isolate as a derivative of the epidemic MRSA (EMRSA)-15 clone, which, as has already been mentioned, has been reported with an increasing frequency from several Portuguese hospitals. The number of VRSA infections reported worldwide is scarce and generally occurs in patients with polymicrobial infections who may be subject to recent use of vancomycin [66].

In Europe, the most prevalent clone of CA-MRSA circulating is ST80 (CC80). Furthermore, different CA-MRSA clones belonging to different STs have been described, including ST1, ST5, ST8, ST22, ST30, ST59, ST80, ST88, ST93 and ST772 (Lakhundi and Zhang, 2018). Several studies conducted in Portugal screened healthy individuals without contact with the hospital environment for the presence of MRSA. A study selected young individuals aged between 13 and 24 years and children from 0 to 5 years of age who attend daycare centers with the aim of evaluating the prevalence, origin and main MRSA clones circulating in the community in Portugal [67]. The results obtained by the authors show that of the 1001 isolates of *S. aureus,* only seven were MRSA (0.7%). Of the seven strains identified, at least five belong to endemic clones from Portuguese hospitals such as the Pediatric, Brazilian and Iberian clones. The low prevalence of MRSA obtained in this study suggests that, in Portugal, MRSA is not widely distributed among the young individuals and children. Another study included 2100 children aged up to 6 years, attending daycare centers. In that study, only three MRSA isolates (0.14%) with typical CA-MRSA characteristics were identified. Furthermore, molecular typing of these isolates showed that they were related to CA-MRSA USA300 and USA700 clones. MRSA USA300 (ST8-t008-IV accessory gene regulator-type I and PVL-positive) is an important CA-MRSA clone epidemic in the United States that has been sporadically found in Europe [68]. In the study by Almeida et al., 3361 individuals aged over 60 years were screened for the presence of MRSA [69]. These individuals are more likely to have one of the risk factors associated with previous contact with MRSA such as hospitalizations or recent surgical interventions. The results obtained are compatible with those previously described. In fact, the prevalence of MRSA obtained among *S. aureus* carriers was 9.2% (62/677) or 1.8% (62/3361) in relation to the total population. Furthermore, most MRSA isolates (82.3%) were associated with clones generally described as HA-MRSA clones, namely the New York/Japan, Pediatric and EMRSA-15 clones and one of the participants from a rural area had ST398 generally associated with LA-MRSA. In addition, among the 62 MRSA isolates, 64.5% were multidrug-resistant and none carried PVL. Similar to previous works, this study suggests that, in Portugal, HA-MRSA clones are no longer restricted to the hospital environment and can be found disseminated in the community. In three of our previous studies, the frequency of *S. aureus* and MRSA was investigated in healthy humans without recent contact with the hospital environment but with direct contact with livestock and pets, namely, cows, donkeys and dogs. Humans handling cows and donkeys were not colonized by MRSA but were S. aureus carriers [70,71]. However, several different clonal lineages were detected among humans, including ST398 and ST97 (animal-associated) and ST30 and ST8 (community-associated). However, 13.3% of the humans in close contact with dogs carried MRSA [72]. Among those isolates, the pediatric clone (ST5-t179-IV) and a variant of USA300 clone lacking the PVL encoding genes were detected.

Colonization and infection by MRSA in animals have been described several times since the 1970s. The occurrence of these cases covers a wide spectrum of animals, ranging from domestic animals to wild species, terrestrial or aquatic, in the natural habitat or in captivity [73,74]. MRSA CC398, which is typically found in livestock, has been identified in humans as a colonizer and a cause of infections. This is a situation that requires rigorous surveillance as it reflects the ability of these bacteria to evolve and adapt to various animal or human hosts. In particular, this clone has the ability to colonize a wide variety of hosts, including dogs, cats, sheep, cows, goats, poultry, rabbits and horses.

### 3.2. MRSA in Pets and Livestock

It has been suggested that LA-MRSA has emerged due to some MRSA adaptation of human origin to the new animal host, which led to the loss of some virulence factors that would be useless in the new environment and, on the other hand, to the acquisition of new mobile genetic elements [75,76]. For this reason, LA-MRSA isolates are genetically distinct from human isolates and currently represent the largest reservoir of MRSA outside hospitals [77]. The spread of zoonotic pathogens between different ecosystems is a high-risk scenario for public health. The implementation of measures to control the global emergence of resistance in these strains will be a key point to stop their spread [78]. The emergence of resistant microorganisms and their genetic spread from animals to humans is due, in part, to the domestication of animals as well as the globalization of the livestock industry. In addition, the continued use of antibiotics in livestock and agricultural activities is another important factor [79]. The spread of microorganisms between animals and humans can occur directly or indirectly. In the first case, this is through direct contact with an infected animal and indirectly through the food chain or by handling contaminated food products of animal origin [80].

People in direct contact with animals such as animal production workers, slaughterhouse workers, livestock transporters and veterinarians, have an increased risk of colonization by LA-MRSA [16,81,82]. In turn, these people can function as a transmission vehicle for other animals and humans. Another form of transmission of LA-MRSA to humans occurs indirectly through the environment. For example, through animal manure, air, contaminate water supplies as well as agricultural crops [83,84].

MRSA CC398 is a pig-adapted LA-MRSA lineage that can be divided into two clades: the classical LA clade and the human clade [85]. It is believed that ST398 was originally a human-associated clone and it has adapted to animals by the loss of integrase group 3 prophages containing the immune evasion cluster (IEC) system genes and with the acquisition of the tetracycline resistance [86,87]. However, it has been shown that a re-adaptation of *S. aureus* CC398 to humans may occur by the acquisition of IEC genes [86,87,88]. MRSA CC398 have been widely documented among humans and animals with ST398-t571 being the most common livestock-associated *S. aureus* lineage in Europe [89]. The first isolation of MRSA CC398 in Portugal was reported in 2009 [90]. This clonal lineage was detected in pigs, and a veterinarian indicated that CC398 can be spread between animals and transmissible from animals to humans. In addition, in this study, MRSA CC30 was isolated in animals and in the environment in a pig farm. Other studies assessing whether healthy animals constitute a reservoir of MRSA in Portugal followed. Regarding the presence of MRSA in swine, studies showed a high frequency of MRSA in pigs and humans in contact with these animals [91]. Furthermore, all isolates belonged to ST398. Therefore, the idea that contact with these animals is a risk factor for MRSA colonization in humans is reinforced. MRSA CC398 has also been reported in calves in Portugal [92]. The MRSA isolates identified carried the *fex*A gene that had already been described in isolates from cattle and swine within a non-conjugative transposon designated Tn558. Thus, an alert was raised regarding the spread of combined resistance to florfenicol and chloramphenicol conferred by the *fex*A gene [93]. In a study conducted on cattle, more than half of the animals (54.3%) were colonized with *S. aureus*, although none of the isolates carried the *mec*A or *mec*C gene [94]. Other studies investigated the presence of MRSA and *S. aureus* in healthy cows in Portugal and also reported the absence of MRSA in cattle [95,96]. It has been suggested that in some cases the contact between the veterinarian and cows is scarce and usually associated with the compulsory surveillance of bovine tuberculosis and brucellosis and do not regularly use antimicrobials in subtherapeutic doses, which may explain the absence of MRSA in Portuguese cattle [95,96]. Nevertheless, other Portuguese studies have reported that 13.1% of the cows screened were colonized by *S. aureus.* The most common *S. aureus* lineage found was ST6-t16615 and most isolates were susceptible to all antimicrobials tested or showed resistance to penicillin [71]. In a study carried out by our research team, 30% of the healthy quails screened carried MRSA and only STs (ST398 and ST6831) and *spa*-types (t011 and t9747) were detected [37]. In that study, all MRSA isolates were multidrug resistant and carried the *tet* genes responsible for tetracycline-resistance which is a marker of LA-MRSA CC398. Two studies conducted with food-factory processing rabbits in Portugal showed that MRSA may be a etiological agent of rabbit lesions [97,98]. The MRSA isolates showed some genetic diversity (ST2855, ST146, ST398 and ST4774), with ST2855 (CC97) being the most predominant clone in both studies. In addition, in one of these studies, human-associated MRSA strains were also detected including isolates belonging to the New York/Japan and EMRSA-15 clones [98].

MRSA infection and colonization in companion animals has been the subject of studies in recent years. Over the years, there has been a significant increase in antimicrobial-resistant and *mec*A-positive isolates. Furthermore, several isolates were multidrug-resistant, which complicates antimicrobial treatment and increases the risk of transfer to humans or from human isolates. Screening studies in companion animals show that MRSA strains typical of hospitals are frequently identified in these animals [99,100]. Currently, it is considered that companion animals can act as reservoirs of important human MRSA clones and contribute to their spread [47]. Couto et al., 2016 characterized the evolution of antimicrobial resistance and the molecular characteristics of 632 *Staphylococcus* spp. isolated from dogs, cats, horses and other companion animals [101]. A total of 11 isolates were MRSA with EMRSA-15 clone as the main MRSA clone detected. In addition, strains belonging to ST398, ST105 and ST5 were also detected. The prevalence of the EMRSA-15 clone in this work is in accordance with previous studies, which demonstrated that there may be a transfer of this clone between humans and companion animals [102]. Coelho et al., 2011 conducted the first study on the detection and molecular characterization of isolates of MRSA strains obtained from healthy dogs in Portugal [103]. A moderate prevalence of MRSA (30%) was detected. Among the 16 positive isolates, 4 different *spa*-types were identified (t032, t432, t747 and t4726) but all isolates belonged to ST22 and SCC*mec* IV (EMRSA-15). Other studies that were conducted in the period between 1999 and 2014, the antimicrobial resistance profiles of bacteria isolated from urinary tract infections (UTI) of dogs and cats were evaluated. The prevalence of *Staphylococcus* spp. was 13.2%. Of these, two isolates belonged to MRSA CC5 (ST5, ST105), a clone that is frequently associated with MRSA of hospital origin in humans.

In the case of horses, it appears that they are infected or colonized by MRSA genetically different from those identified in pets. In this context, ST1, ST22, ST254 and ST398 has been the most prevalent clone in cases of colonization and infection in horses in Europe [104]. Couto et al. sampled Lusitanian horses admitted to a veterinary hospital and the samples were screened for MRSA [105]. The results obtained show that the prevalence of MRSA colonization was 3% (2/71). After molecular characterization, one of the MRSA isolates was found to be associated with the Pediatric (ST5-IV/VI) and the New York/Japan (ST5-II) clones. It is important to note that none of these clones had yet been detected in equine samples in Europe. The second MRSA isolate was identified as ST398, the main clone colonizing horses admitted to veterinary hospitals in Europe, according to previous studies [106].

## 4. Conclusions

MRSA is a ubiquitous microorganism that has a remarkable ability to adapt to different environments and hosts, giving rise to successful epidemic strains that make it continue to be a major threat to public health. The convergence of habitats has made contact between humans, pets, livestock and wild animals increasingly common. Over the years, there have been several shifts in the MRSA clones that circulated in Portugal. However, for more than a decade, the predominant clone in both the hospital and community settings has been EMRSA-15. In fact, the most recently published studies show that this clone continues to be the most frequently isolated and with high prevalence when compared with other endemic clones from Portugal. As for animals, both pets and livestock, the scenario is the same in Portugal as in the rest of Europe where the dominant clonal lineages in pets are the same as in humans, and it is MRSA CC398 in livestock. Therefore, better implementation of measures is also needed in the future to control zoonotic MRSA reservoirs and limit the global spread.

## Figures and Tables

**Figure 1 pathogens-11-01110-f001:**
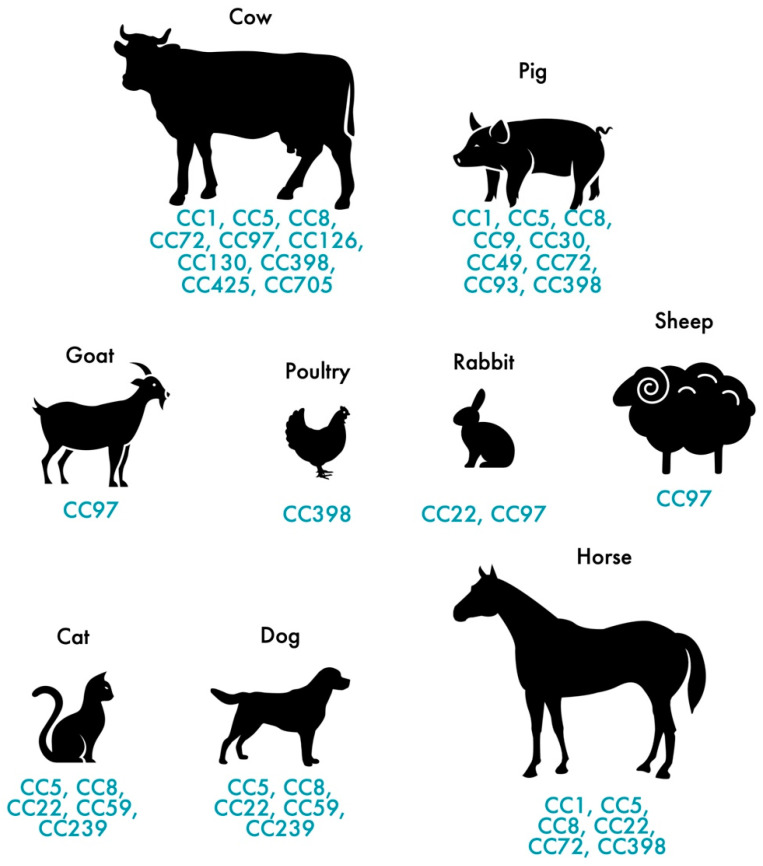
Distribution of the most frequent MRSA and MSSA clones in pets and livestock.

**Figure 2 pathogens-11-01110-f002:**
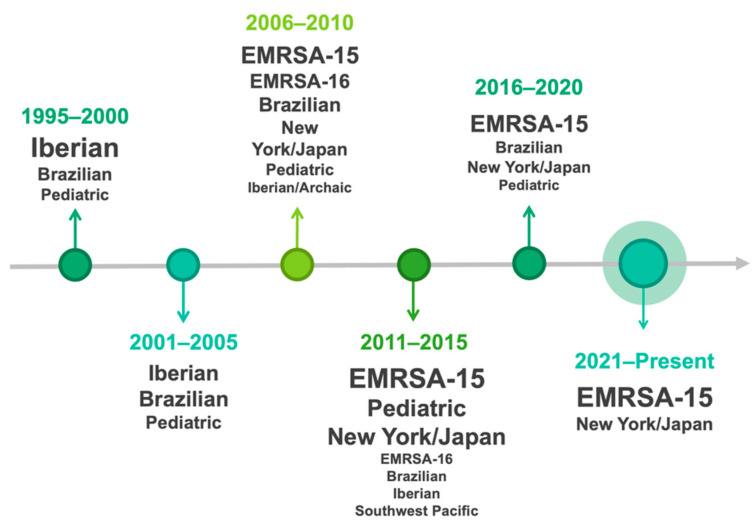
The shift of MRSA clonal strains’ prevalence over the years in humans in Portugal.

**Table 1 pathogens-11-01110-t001:** Pandemic MRSA lineages and their worldwide distribution [27,28].

Clonal Complex (CC)	Sequence Type (ST)	*Spa*-Type	SCC*mec* Type	Clone	Geographic Distribution
CC1	ST6	t304 and variants	IVa	Middle East	The Middle East and Europe
CC5	ST5	t001, t002, t003, t010, t045, t053, t062, t105, t178, t179, t187, t214, t311, t319, t389, t443	II	New York/Japan, USA100	United States, Japan, Europe, Australia and South Korea
		t001, t002, t003, t010, t045, t053, t062, t105, t178, t179, t187, t214, t311, t319, t389, t443	IV	Pediatric/USA800	South America and Europe
CC8	ST239	t030, t037, t234, t387, t388	III	Brazilian/Hungarian	Europe, South America, Asia and Africa
	ST247	t008, t051, t052, t054, t200	I	Iberian/EMRSA-5	Europe and the United States
	ST250	t008, t009, t194	I	Archaic	Worldwide
CC22	ST22	t005, t022, t032, t223, t309, t310, t417, t420	IV	EMRSA-15	Europe, Australia and Canada
CC30	ST36	t018, t253, t418, t419	II	EMRSA-16, USA200	Europe, North America, and Australia
	ST30	t012, t019, t1143, t300	IV	USA1100/South West Pacific	America, Australia and the Western Pacific
CC45	ST45	t004, t015, t026, t031, t038, t050, t065, t204, t230, t390	IV	Berlin, USA600	Europe and the United States
ST80	ST80	t044, t203, t131, t1028, t1200	IV	European	Europe, North Africa and the Middle East
	ST93	t3949 t202 t15361 t4699 t17089 t16949 t17272	IV	Queensland	Australia

Abbreviations. ST: sequence type; CC: clonal complex; SCC*mec*: Staphylococcal Cassette Chromosome *mec.*

## Data Availability

Not applicable.

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
