# Peer review of "MRSA in Humans, Pets and Livestock in Portugal: Where We Came from and Where We Are Going"

_pathogens, 2022, doi:10.3390/pathogens11101110_

Round 1

Reviewer 1 Report

The manuscript is interesting and well organized, there are no corrections to be made, I just wonder why there are two sentences written in bold.

Author Response

The manuscript is interesting and well organized, there are no corrections to be made, I just wonder why there are two sentences written in bold.

A: We appreciate the reviewer’s comments. Altered in the new version of the manuscript.

Reviewer 2 Report

The authors attempted to do a literature review on MRSA in Portugal, covering humans, pets, and livestock. However, the structure and conclusion of the whole article are not very appropriate. Here are some main issues and some minor issues: 

1. The statement in line 25 of the abstract is not based on the results of solid evidence. It is generally believed that hospital MRSA and community MRSA have some different characteristics, and over time, community MRSA evolved and entered the hospital. However, most of the nosocomial MRSA found in the community was because the patient had a relevant medical history in the past year, as described in Klevens, R. M. et al. JAMA 2007;298:1763-1771. Therefore, it is best to have a rigorous definition of the source of the patient strain in such studies in order to make more appropriate conclusions. 

2. To investigate the phenomenon of clone substitution, it is best to have national multi-centers regularly collect strains and analyze them according to the same practice.

3. When the author introduces the clone of strain, they should integrate the literature to describe it. For example, lines 106-113 show that both paragraphs describe the "New York/Japan" clone.

4. Line 89 should be Archaic (ST250-I).

5. The sentence on line 103 says something weird.

6. On line 238 there are a lot of words that need to be italicized.

Author Response

The authors attempted to do a literature review on MRSA in Portugal, covering humans, pets, and livestock. However, the structure and conclusion of the whole article are not very appropriate. Here are some main issues and some minor issues: 

 1. The statement in line 25 of the abstract is not based on the results of solid evidence. It is generally believed that hospital MRSA and community MRSA have some different characteristics, and over time, community MRSA evolved and entered the hospital. However, most of the nosocomial MRSA found in the community was because the patient had a relevant medical history in the past year, as described in Klevens, R. M. et al. JAMA 2007;298:1763-1771. Therefore, it is best to have a rigorous definition of the source of the patient strain in such studies in order to make more appropriate conclusions. 

A: We agree with the reviewer comment. Altered in the new version of the manuscript.

2. To investigate the phenomenon of clone substitution, it is best to have national multi-centers regularly collect strains and analyze them according to the same practice.

A: We totally agree with the reviewer’s comment. Nevertheless, and although it is possible to state that there has been a clone substitution, it is possible to observe through the available literature that some clones that were more frequent in the past are no longer so frequent and that others that were less frequent in the past are now identified in a higher percentage.

3. When the author introduces the clone of strain, they should integrate the literature to describe it. For example, lines 106-113 show that both paragraphs describe the "New York/Japan" clone.

A: Altered in the new version of the manuscript.

4. Line 89 should be Archaic (ST250-I).

A: Altered in the new version of the manuscript.

5. The sentence on line 103 says something weird.

A: Altered in the new version of the manuscript.

6. On line 238 there are a lot of words that need to be italicized.

A: Altered in the new version of the manuscript.

Reviewer 3 Report

This is an interesting review of the occurrence of MRSA strains in humans, pets and livestock in Portugal with particular attention to the emergence and evolution of different MRSA clones.  The subject matter is still relevant in the context of epidemiological, therapeutic and clinical problems. The manuscript is well written, but some elements should be revised by introducing additions related to the main topic.

-          Sections: “Abstract” and “Introduction”, need to be supplemented with the purpose of the publication

-          When describing the occurrence of MRSA strains in humans, pets and livestock, it is extremely important to pay attention to drug resistance of these strains. Most of the therapeutic, clinical and epidemiological problems stem from it. This aspect was marginalized by the authors of the manuscript. Some parts of the text need to be supplemented with information on antibiotic resistance in MRSA strains.

-          Line number 221 - Were the main clones associated with hospital in Portugal multi-drug resistant? To what antibiotics they showed resistance? -          Line number 237 - Whether vancomycin resistance is present in MRSA strains in Portugal? Have such strains been reported among patients in Portugal?

-          Line number 279 - What is the antibiotic-resistance of MRSA strains isolated from humans in close contact with animals? Are they multi-drug resistant strains?

Author Response

This is an interesting review of the occurrence of MRSA strains in humans, pets and livestock in Portugal with particular attention to the emergence and evolution of different MRSA clones.  The subject matter is still relevant in the context of epidemiological, therapeutic and clinical problems. The manuscript is well written, but some elements should be revised by introducing additions related to the main topic.

A: We appreciate the reviewer’s comments.

-          Sections: “Abstract” and “Introduction”, need to be supplemented with the purpose of the publication

A: Altered in the new version of the manuscript.

-          When describing the occurrence of MRSA strains in humans, pets and livestock, it is extremely important to pay attention to drug resistance of these strains. Most of the therapeutic, clinical and epidemiological problems stem from it. This aspect was marginalized by the authors of the manuscript. Some parts of the text need to be supplemented with information on antibiotic resistance in MRSA strains.

A: We agree with the reviewers that the drug resistance of the strains is, in fact, extremely important. Nevertheless, the main objective of the review is to report the clones that circulate among humans and among animals in Portugal and to try to understand if clone shifting has occurred over the years. In addition, some of the studies do not report in detail the phenotypic and genotypic resistances of the respective isolates, which makes comparison more difficult.

-          Line number 221 - Were the main clones associated with hospital in Portugal multi-drug resistant? To what antibiotics they showed resistance? -          Line number 237 - Whether vancomycin resistance is present in MRSA strains in Portugal? Have such strains been reported among patients in Portugal?

A: More information was added to this part.

-          Line number 279 - What is the antibiotic-resistance of MRSA strains isolated from humans in close contact with animals? Are they multi-drug resistant strains?

A: More information was added to this part.

Reviewer 4 Report

The review entitled by Silva et al., 2022 reports a good overview of the bibliography about MRSA in Europe, with a special focus on Portugal.

The paper is well written, it is easy to follow and the bibliography is comprehensive.

I think there are a couple of major points to be addressed before publication (which I advocate openly).

11. Readers might not be familiar with naming/classification of MRSA. I am personally familiar with the concepts of CC and STs, but there is nowhere mention of how these are identified, or the meaning of “spa” type and how this is classified. Also, the authors refer to a variety of “common names” (ie Pediatric). I think it would be beneficial for the reader to have this specified somewhere

    2.The authors divide the paper into 2 blocks: a general part and a Portugal-specific section. However, It seems to me that many “general” contents are reported in the Portugal specific part. Please rearrange accordingly.

Some minor comments/typos:

Line 19: use pathogenic instead of disease producing

Line 39: S. aureus missing before the brackets

Line 41-44: you repeat twice the same thing before and after the comma

Line 77: characterization instead of differentiation (used 2x)

Line 90: extra space before NY

In general: keep consistent how you refer to the strains, with or without “”

Line 96: first time mentioning STs, you should write Sequence Types (STs)

Tab1: check the justification of wells, I would add some horizontal lines to separate better

Line 102-3: I would make it clear here that Iberian is synonym od EMRSA-15

Line 126: checjk that 20-90% is correct as it seems to me it is a big fork.

Line 137: missing space before generally

Fig1: please align the text under the animals

In general, CC stands for clonal complex and not clones

Fig 2: I guess the font is referred to the prevalence in the years, if so you should specify this

In general, keep consistent the thousands ( nothing vs a space)

Line 231: first trime mentioning VRSA: please specify the full meaning.

In general: S. aureus must be italics

Line 259: what is agr type?

Line 330: on cattle

Line 344: food factory processing rabbits

 Line 385: delete organism

Author Response

The review entitled by Silva et al., 2022 reports a good overview of the bibliography about MRSA in Europe, with a special focus on Portugal.

The paper is well written, it is easy to follow and the bibliography is comprehensive.

A: We appreciate the reviewer’s comments.

I think there are a couple of major points to be addressed before publication (which I advocate openly).

1. Readers might not be familiar with naming/classification of MRSA. I am personally familiar with the concepts of CC and STs, but there is nowhere mention of how these are identified, or the meaning of “spa” type and how this is classified. Also, the authors refer to a variety of “common names” (ie Pediatric). I think it would be beneficial for the reader to have this specified somewhere

A: We agree with the reviewer. Altered in the new version of the manuscript.

    2.The authors divide the paper into 2 blocks: a general part and a Portugal-specific section. However, It seems to me that many “general” contents are reported in the Portugal specific part. Please rearrange accordingly.

A: The text was rearranged accordingly.

Some minor comments/typos:

Line 19: use pathogenic instead of disease producing

A: Altered in the new version of the manuscript.

Line 39: S. aureus missing before the brackets

A: Altered in the new version of the manuscript.

Line 41-44: you repeat twice the same thing before and after the comma

A: Altered in the new version of the manuscript.

Line 77: characterization instead of differentiation (used 2x)

A: Altered in the new version of the manuscript.

Line 90: extra space before NY

A: Altered in the new version of the manuscript.

In general: keep consistent how you refer to the strains, with or without “”

A: Altered in the new version of the manuscript.

Line 96: first time mentioning STs, you should write Sequence Types (STs)

A: Altered in the new version of the manuscript.

Tab1: check the justification of wells, I would add some horizontal lines to separate better

A: Altered in the new version of the manuscript.

Line 102-3: I would make it clear here that Iberian is synonym od EMRSA-15

A: Altered in the new version of the manuscript.

Line 126: checjk that 20-90% is correct as it seems to me it is a big fork.

A: Altered in the new version of the manuscript.

Line 137: missing space before generally

A: Altered in the new version of the manuscript.

Fig1: please align the text under the animals

Fig 2: I guess the font is referred to the prevalence in the years, if so you should specify this

A: Altered in the new version of the manuscript.

In general, keep consistent the thousands ( nothing vs a space)

Line 231: first trime mentioning VRSA: please specify the full meaning.

A: Altered in the new version of the manuscript.

In general: S. aureus must be italics

A: Altered in the new version of the manuscript.

Line 259: what is agr type?

A: Altered in the new version of the manuscript.

Line 330: on cattle

A: Altered in the new version of the manuscript.

Line 344: food factory processing rabbits

A: Altered in the new version of the manuscript.

Line 385: delete organism

A: Altered in the new version of the manuscript.

Round 2

Reviewer 2 Report

I have no other questions. But please check the initial abbreviation position change due to the addition.